# Attention-Based Bi-Prediction Network for Versatile Video Coding (VVC) over 5G Network

**DOI:** 10.3390/s23052631

**Published:** 2023-02-27

**Authors:** Young-Ju Choi, Young-Woon Lee, Jongho Kim, Se Yoon Jeong, Jin Soo Choi, Byung-Gyu Kim

**Affiliations:** 1Department of IT Engineering, Sookmyung Women’s University, Seoul 04310, Republic of Korea; 2Department of Computer Engineering, Sunmoon University, Asan 31460, Republic of Korea; 3Media Coding Research Section, Electronics and Telecommunications Research Institute, Daejeon 34129, Republic of Korea

**Keywords:** 5G, versatile video coding, attention mechanism, bi-prediction, convolutional neural network

## Abstract

As the demands of various network-dependent services such as Internet of things (IoT)
applications, autonomous driving, and augmented and virtual reality (AR/VR) increase, the fifthgeneration
(5G) network is expected to become a key communication technology. The latest video
coding standard, versatile video coding (VVC), can contribute to providing high-quality services by
achieving superior compression performance. In video coding, inter bi-prediction serves to improve
the coding efficiency significantly by producing a precise fused prediction block. Although block-wise
methods, such as bi-prediction with CU-level weight (BCW), are applied in VVC, it is still difficult for
the linear fusion-based strategy to represent diverse pixel variations inside a block. In addition, a
pixel-wise method called bi-directional optical flow (BDOF) has been proposed to refine bi-prediction
block. However, the non-linear optical flow equation in BDOF mode is applied under assumptions,
so this method is still unable to accurately compensate various kinds of bi-prediction blocks. In
this paper, we propose an attention-based bi-prediction network (ABPN) to substitute for the whole
existing bi-prediction methods. The proposed ABPN is designed to learn efficient representations of
the fused features by utilizing an attention mechanism. Furthermore, the knowledge distillation (KD)-
based approach is employed to compress the size of the proposed network while keeping comparable
output as the large model. The proposed ABPN is integrated into the VTM-11.0 NNVC-1.0 standard
reference software. When compared with VTM anchor, it is verified that the BD-rate reduction of the
lightweighted ABPN can be up to 5.89% and 4.91% on Y component under random access (RA) and
low delay B (LDB), respectively.

## 1. Introduction

The appearance of the fifth-generation (5G) network brings about technological innovation for media services in wireless systems such as Internet of things (IoT) applications, autonomous driving, and augmented and virtual reality (AR/VR) services [1,2,3,4,5]. In accordance with support of an ultrahigh-speed data transmission, the quality of service can be improved. With the development of the display industry, the aforementioned media services are expected to rely on continued video evolution toward 8K resolutions [6]. The 5G technology includes the following four typical characteristics: high speed, low delay, large capacity, and mobility [7]. To achieve these features for high bit rate video, the demand for high-performance video compression technology has increased exponentially.

After successful standardization of the previous video coding standards, e.g., advanced video coding (H.264/AVC) [8] and high-efficiency video coding (H.265/HEVC) [9], versatile video coding (VVC) [10] has been finalized by the joint video exports team (JVET) of ITU-T Video Coding Experts Group (VCEG) and ISO/IEC Moving Picture Experts Group (MPEG) in July 2020. The VVC has been designed to provide a significant bit rate reduction compared to HEVC. In addition, VVC is expected to be utilized not only for high-resolution video such as ultrahigh-definition (UHD) video, but also in various types of video content and applications, e.g., high dynamic range (HDR), screen content, online gaming applications, 360° video for immersive AR/VR. Therefore, VVC is the most suitable video coding standard for media services over 5G networks.

As with existing video coding standards, VVC also follows the classic block-based hybrid video coding framework consisting of several core modules. In the hybrid coding scheme of VVC, the quadtree with a nested multitype tree using binary and ternary splits (QT-MTT) block partitioning structure has been applied to split a picture into different types of blocks [10]. The QT-MTT block partitioning has been designed to support more flexible shapes and larger block sizes than the structure in HEVC. The coding unit (CU) is a basic unit for signaling prediction information. For each CU, intra prediction and/or inter prediction is performed, followed by transform, quantization, and entropy coding. Furthermore, the in-loop filtering process improves the quality of the reconstructed frame.

The core of video coding is removing redundancy inherent in video signals to the extent that it is not perceived visually. One of the most important processes in video coding is the prediction process that finds pixel-based redundant information, sets it as the prediction signals, and removes the predictable signals. Intra prediction removes spatial redundancy between adjacent pixels or blocks within a frame. Inter prediction removes temporal redundancy with blocks in previous and/or future neighboring frames. In particular, due to the characteristics of the video, there is a high probability that a block similar to the block currently being coded exists in adjacent frames. For this reason, inter prediction mode contributes more significantly to overall coding efficiency than intra prediction mode. Inter prediction methods in VVC can be classified into the whole block-based inter prediction [11] and the subblock-based inter prediction [12]. Furthermore, in both whole block-based and subblock-based inter prediction schemes, adaptive motion vector prediction (AMVP) mode and merge mode are performed.

Two kinds of inter prediction modes are allowed in video coding standards, namely uni-prediction and bi-prediction.Compared to the uni-prediction mode that utilizes one prediction block, the bi-prediction mode that combines the signals of two prediction blocks makes a more accurate prediction block. The precise prediction signal generation results in improved coding efficiency. Taking four sequences with varying the resolution and the degree of motion, we investigate the ratio of selected modes for whole CUs in B-slices in VVC test model VTM-11.0 NNVC-1.0 [13] in Figure 1. As shown in Figure 1, in all results, inter bi-prediction mode occupies the largest proportion, followed by inter uni-prediction mode and intra mode. Generally, the smoother the motion change like camera moving, the more advantageous it is for bi-prediction.For this reason, particularly, in sequences *BQTerrace* and *BQSquare* containing stable motions, bi-prediction mode is observed by a heavy percentage. This investigation demonstrates that the performance of bi-prediction method contributes to the overall coding efficiency.

In VVC, the bi-prediction block is generated by utilizing three bi-prediction modes. The bi-prediction modes consist of simple averaging, a weighted averaging called BCW (bi-prediction with CU-level weight), and averaging with bi-directional optical flow (BDOF)-based refinement. In the averaging mode, two reference blocks are linearly combined by average function as with the bi-prediction mode of HEVC. In VVC, the BCW mode and the BDOF mode are newly adopted to enhance the accuracy of bi-prediction block generation. The BCW mode is the extended method of simple averaging to allow weighted averaging of the two prediction signals. The BDOF mode is the pixel-wise refinement method to compensate the precise motion missed by the block-based motion compensation based on the 4×4 sub-block.The BDOF samples of the CU are calculated under several assumptions. The assumptions are that the luminance of objects is constant according to the optical flow, the objects are moving with constant speed, and the motion with the surrounding sample is the same.

Compared with simple averaging bi-prediction mode, the coding efficiency can be substantially improved by enhancing the prediction accuracy with the strengthened methods. However, this linear fusion-based approach such as the BCW mode may still have limitations in representing diverse motion variations. In the majority of natural circumstances, the changes of pixels in one block may be inconsistent. The linear function-based imprecise fusion for irregular variations can bring a lot of residual signals. In addition, the BDOF mode is still limited to obtain precise prediction block when the actual motion deviates from the aforementioned assumptions of optical flow. In addition, the BDOF mode aims to adjust the bi-prediction samples finely based on sub-CU with several conditions and assumptions. Hence, this strategy is still limited to obtain precise prediction block when the actual motion deviates from the aforementioned assumptions of optical flow.

Recently, the enhancement accuracy in various low-level tasks [14,15,16,17] has improved significantly by learning the non-linear patch-to-patch mapping function directly with the convolutional neural network (CNN). In the field of video coding, in particular, many researchers have been explored CNN-based in-loop filtering [18,19,20] and post-filtering [21,22,23] tasks actively. Aside from these, a number of CNN-based studies in video coding tasks, such as intra prediction [24,25] and inter prediction [26,27] have been proposed.

Until now, the deep learning-based works for inter bi-prediction [28,29] aimed to increase the performance of traditional bi-prediction in HEVC. By applying BCW and BDOF, the bi-prediction module of VVC was improved compared with that of HEVC. Because the target anchor has been strengthened, the existing simple CNN network-based strategy has a limitation to substitute the bi-prediction process of VVC completely. Hence, to replace all three bi-prediction modes of VVC, a more complex and advanced network is necessary.

In this paper, we propose an attention-based bi-prediction network (ABPN) to generate a fine bi-prediction block as a replacement of bi-prediction methods in VVC. The proposed ABPN can extract an elaborate fused feature by generating attention map between two reference blocks. In addition, the proposed network is designed by utilizing a local and global skip-connection structure. The stacked residual blocks [30] take a role of local skip-connection blocks and the global skip-connection is structured by adding the predicted residual block to a block averaging two input blocks. This architecture makes it possible to construct a deeper network. Furthermore, we adopt knowledge distillation (KD) [31] to make the proposed ABPN a lightweighted network effectively. The major contributions of this paper are summarized as follows.

We propose an attention-based bi-prediction network (ABPN). Different from the existing bi-prediction methods in VVC, the proposed method can enhance the quality of bi-prediction block by using a CNN-based manner. Because the proposed ABPN can reduce the bit rate while providing higher visual quality compared to the original VVC, the efficiency of transmission over 5G networks can be increased considerably.The depth of the proposed network is deeper than that of the networks proposed in existing deep learning-based bi-prediction studies. Because there are more bi-prediction modes of VVC than HEVC, it is an mandatory choice for replacing all modes. In this paper, we utilize a learning technique named KD which distills the knowledge from a larger deep neural network into a small network. It allows the number of parameters to be reduced while keeping the quality of the result similarly.The proposed ABPN is integrated into VTM-11.0 NNVC-1.0 anchor on JVET neural network-based video coding (NNVC) standard. The experimental results demonstrate that the proposed method achieves superior coding performance compared with the VTM anchor.

The rest of the paper is organized as follows. In Section 2, we introduce the related works. In Section 3, we present the proposed methodology. The experimental results and discussions are shown in Section 4. Finally, Section 5 makes the concluding remarks for this paper.

## 2. Related Works

### 2.1. Traditional Bi-Prediction Method in Video Coding

The weighted prediction (WP) is a global compensation method in frame-level to efficiently deal with brightness variation. In H.264/AVC and H.265/HEVC standards, the WP coding tool is applied with the weighting parameters of reference pictures. Even in VVC standard, WP method is supported to compensate the inter prediction signal. The WP method in VVC allows weight and offset to be signalled for each reference picture in each of the reference picture lists L0 and L1. Although there are global variations for the whole frame in video contents, the irregular local changes also exist. Therefore, block-level compensation of the inter prediction signal is required to improve the accuracy of the prediction.

In VVC, a novel weighted bi-prediction method for CU-level, named bi-prediction with CU-level weight (BCW), and a refinement method with bi-directional optical flow (BDOF) were applied. In contrast with HEVC, VVC adopted the block-based blending method for increasing the prediction precision by using the BCW scheme. For every CU, the BCW method is performed with several candidate weights. For the low-delay pictures, five weights are used, and three weights are used otherwise. To refine the bi-prediction signal of a CU, the BDOF method is applied. For each 4 × 4 sub-CU, a motion refinement is calculated by minimizing the difference between the L0 and L1 prediction blocks. Then, the motion refinement is utilized to adjust the bi-predicted signal. In order to avoid interactions between frame-based and block-based bi-prediction methods, the BCW and the BDOF modes are not used if the WP mode is used.

### 2.2. Deep Learning-Based Inter Prediction in Video Coding

Table 1 shows the problem formulation and related methods for deep learning-based inter prediction in video coding. Inspired by the success of deep learning in many computer vision tasks, numerous deep learning-based research projects for video coding have begun. To refine and/or replace the traditional inter prediction methods in video coding standards, some works have been proposed. Huo et al. [32] proposed a CNN-based motion compensation refinement (CNNMCR) scheme to refine the prediction block for uni-prediction in HEVC. Wang et al. [33] proposed a uni-prediction refinement network consisting of a fully connected network (FCN) and a CNN in HEVC which was named as a neural network-based inter prediction (NNIP) algorithm.

Furthermore, there are several studies related to the extrapolation and interpolation of the frame, which can be used as an additional reference frame. Zhao et al. [34] proposed a deep virtual reference frame enhancement CNN model (VECNN) to replace the traditional frame rate up conversion (FRUC) algorithm in both frame and coding tree unit (CTU) level. Liu et al. [35] proposed a multi-scale quality attentive factorized kernel convolutional neural network (MQ-FKCNN) to generate additional reference frames. Huo et al. [36] have proposed a deep network-based frame extrapolation method using reference frame alignment for HEVC and VVC. Recently, Choi et al. [37] proposed a neural network-based frame estimation from two reference frames by applying an affine transformation-based scheme.

For improvement of bi-prediction in HEVC, Zhao et al. [28] proposed a CNN-based bi-prediction scheme based on a patch-to-patch inference strategy. The proposed network in [28] stacked six convolution layers with skip connection. For the luma components of prediction unit (PU) with the sizes 16×16, 32×32, and 64×64, this network replaces traditional averaging bi-prediction method. Later on, Mao et al. [29] proposed a CNN-based bi-prediction method utilizing both spatial neighboring regions and temporal display orders as extra inputs to further improve the prediction accuracy which named as STCNN. By applying correlation of neighboring pixels and video frames, this work replaced averaging bi-prediction method effectively. As in [28], the proposed network in [29] applied six convolution layers with skip connection. Furthermore, as an additional experiment, this method replaced the bi-prediction mode in HEVC by combining the BIO [38] method. However, the target of this method is still the replacement of averaging bi-prediction method.

As mentioned in the above, both works aimed at replacement of an average function-based bi-prediction method in HEVC. Because the BCW and the BDOF methods are supported for bi-prediction including the average method in VVC, it is more difficult to replace the traditional bi-prediction.In other words, the bi-prediction process in VVC is more complicated and improved than that of HEVC because these novel bi-prediction tools lead to significant performance gain. Therefore, in order to replace the enhanced bi-prediction process, it is necessary to construct a more elaborate CNN-based framework.

## 3. Methodology

In this section, the proposed attention-based bi-prediction network (ABPN) is presented. First, the architecture of the proposed network is described. Secondly, the knowledge distillation-based training technique for network lightning is illustrated. Finally, the details of integrating the proposed ABPN into VVC will be described.

### 3.1. Architecture of the Proposed ABPN

In contrast to the existing bi-prediction methods in VVC, the proposed method can enhance the quality of bi-prediction block by using a CNN-based manner. Because the BCW and the BDOF methods are supported for bi-prediction including the average method in VVC, it is more difficult to replace the traditional bi-prediction.The proposed ABPN which is designed as a sophisticated structure has an advantage in a high performance video codec such as VVC.

The proposed network is illustrated in Figure 2. Given two reference prediction blocks $P_{0}$ and $P_{1}$, where *H* is height and *W* is width, the goal of the network is to estimate the bi-prediction block $P_{bi}$. The output block size of the network is H×W equal to the size of the input blocks.

Two reference prediction blocks are results obtained during each motion-estimation process in different or the same reference frames. In motion-compensation process, one bi-prediction block is produced by blending two motion-compensated blocks after performing uni-prediction mode. The essential part of the fusion of two prediction blocks is finding accurate intermediate motion information between motions of two prediction blocks. Meanwhile, the important edge and texture details in each reference prediction blocks should be maintained. By utilizing the attention map between two reference features, we can obtain appropriate motion information for the current target block. To retain low-frequency and high-frequency characteristics that should not be missed, we exploit the local and global residual learning structures. Each of the two input prediction blocks are fed into three CNN layers to increase the number of features,
(1)$$\begin{array}{c} F_{0}^{1}=f\left(W_{0}^{1}*P_{0}+b_{0}^{1}\right), \end{array}$$
(2)$$\begin{array}{c} F_{0}^{2}=f\left(W_{0}^{2}*F_{0}^{1}+b_{0}^{2}\right), \end{array}$$
(3)$$\begin{array}{c} F_{0}^{3}=f\left(W_{0}^{3}*F_{0}^{2}+b_{0}^{3}\right), \end{array}$$
(4)$$\begin{array}{c} F_{1}^{1}=f\left(W_{1}^{1}*P_{1}+b_{1}^{1}\right), \end{array}$$
(5)$$\begin{array}{c} F_{1}^{2}=f\left(W_{1}^{2}*F_{1}^{1}+b_{1}^{2}\right), \end{array}$$
(6)$$\begin{array}{c} F_{1}^{3}=f\left(W_{1}^{3}*F_{1}^{2}+b_{1}^{3}\right), \end{array}$$
where f(·), $W_{i}^{j}$, $F_{i}^{j}$, and $b_{i}^{j}$ denote the activation function, the weight, the output feature, and the bias of *j*th convolution layer, respectively, and *i* is the index of the reference block. For activation function of all convolution layers except the last layer in the overall network, the leaky rectifier linear unit (LeakyReLU) is used. The rectifier linear unit (ReLU) activation function is commonly applied in CNN. However, if the output value of the convolution layer is a negative value, the output value is converted to zero. In training, a zero value has a bad effect. The LeakyReLU function which multiplies 0.01 to negative value can resolve this problem. The attention map between two features $F_{0}^{3}$ and $F_{1}^{3}$ can be obtained by using the dot product and the sigmoid activation function,
(7)$$\begin{array}{c} Att=Sigmoid(F_{0}^{3}\cdot F_{1}^{3}), \end{array}$$
where · and Sigmoid(·) denote the dot product and the sigmoid activation function, respectively. The sigmoid activation function converts the output values to the values in the range of [0,1]. As opposed to the softmax function which is more appropriate for inferring a class vector based on the probability, the sigmoid function is more suitable in generating an attention map. After that, the attention map is then multiplied to the embedded features obtained by first convolution layers,
(8)$$\begin{array}{c} F_{0}^{a}=F_{0}^{1}⊙Att, \end{array}$$
(9)$$\begin{array}{c} F_{1}^{a}=F_{1}^{1}⊙Att, \end{array}$$
where ⊙ denotes the element-wise product, $F_{0}^{a}$ and $F_{1}^{a}$ are the attended features of two reference blocks. The process of generating attended features is illustrated with blue lines in Figure 2. The embedded features and attended features are then concatenated:(10)$$\begin{array}{c} F^{c}=\left[F_{0}^{3},F_{1}^{3},F_{0}^{a},F_{1}^{a}\right], \end{array}$$
where [,,,] denotes the concatenation function, and $F^{c}$ is the concatenated feature. The concatenated feature passes through two CNN layers: (11)$$\begin{array}{c} F^{4}=f\left(W^{4}*F^{c}+b^{4}\right), \end{array}$$(12)$$\begin{array}{c} F^{5}=f\left(W^{5}*F^{4}+b^{5}\right). \end{array}$$

After that, $F^{5}$ is fed into $N_{rb}$ residual blocks wherein there are no batch normalization units [30]. The stacked residual blocks play a role as local skip-connection blocks. Each residual block can be expressed as
(13)$$\begin{array}{c} F_{rb(k)}^{1}=f\left(W_{rb(k)}^{1}*F_{rb(k-1)}^{out}+b_{rb(k)}^{1}\right), \end{array}$$
(14)$$\begin{array}{c} F_{rb(k)}^{2}=W_{rb(k)}^{2}*F_{rb(k)}^{1}+b_{rb(k)}^{2}, \end{array}$$
(15)$$\begin{array}{c} F_{rb(k)}^{out}=F_{rb(k)}^{2}+F_{rb(k-1)}^{out}, \end{array}$$
where $W_{rb(k)}^{l}$, $F_{rb(k)}^{l}$, and $b_{rb(k)}^{l}$ denote the weight, the output feature, and the bias of the *l*th convolution layer in the *k*th residual block, respectively. At this time, the input feature of first residual block $F_{rb(0)}^{l}$ equals $F^{5}$. Then, the output feature of the last residual block $F_{rb(N_{rb})}^{2}$ passes through two CNN layers: (16)$$\begin{array}{c} F^{6}=f\left(W^{6}*F_{rb(N_{rb})}^{2}+b^{6}\right), \end{array}$$(17)$$\begin{array}{c} F^{7}=W^{7}*F^{6}+b^{7}. \end{array}$$

Finally, the final bi-prediction block is obtained by adding the predicted residual block to a block averaging two input blocks,
(18)$$\begin{array}{c} P_{bi}=F^{7}+Average\left(P_{0},P_{1}\right), \end{array}$$
where Average() denotes the average function. This global skip-connection process is illustrated with green lines in Figure 2. The proposed network can preserve important features of diverse depth of network by using local and global residual learning.

### 3.2. KD-Based Lightweighted Network Design

In the majority of cases, deep learning-based prediction methods greatly increase encoding and decoding computational complexity as much as they improve coding performance. In particular, in encoder, because a prediction process is performed based on CU-level, every CU in the recursive block partitioning structure should be encoded for all of the candidate prediction modes. Due to these various numbers of cases, computational complexity increasing on deep learning-based prediction mode is unavoidable. Moreover, from the decoder point of view it is important to deploy deep learning models on devices with limited resources. However, a simple network for prediction method can cause insufficient results.

To compress the size of network effectively, we adopt the knowledge distillation (KD) [31] strategy in the training stage. The KD-based training structure in the deep learning network is a teacher–student architecture. Once the massive model has been trained, this large model becomes a teacher model. To obtain an optimal lightweight model, the teacher model transfers the knowledge to small student model. Through the KD mechanism, we can acquire a more suitable model in limited circumstance.

Because the KD has been initially proposed for the recognition task [31], we design the loss function for the low-level vision task. The structure with the KD-based learning strategy is illustrated in Figure 3. Two input blocks are fed into both teacher model and student model. The teacher model is a pretrained large model, and the student model is one to be trained.

In this paper, we compress the network in two respects—the number of features of all CNN layers except the last layer, and the number of residual blocks. When the number of features and residual blocks of the teacher model are defined as $N_{f\_t}$ and $N_{rb\_t}$, the aims of KD-based training discover the proper number of features and residual blocks of student model, $N_{f\_s}$ and $N_{rb\_s}$. For that, the output of teacher model is utilized in the loss function, and only the weights of the student model are updated. We design a loss function consisting of the distillation loss $L_{d}$ and the student loss $L_{s}$. For both losses, we use the Charbonnier penalty function [39],
(19)$$\begin{array}{c} L_{d}=\sqrt{{∥P_{bi\_t}-P_{bi\_s}∥}^{2}+\varepsilon^{2}}, \end{array}$$
(20)$$\begin{array}{c} L_{s}=\sqrt{{∥\hat{P_{bi}}-P_{bi\_s}∥}^{2}+\varepsilon^{2}}, \end{array}$$
where $P_{bi\_t}$, $P_{bi\_s}$, and $\hat{P_{bi}}$ denote the prediction output of teacher model, student model, and ground truth, respectively, and ε set to $1\times 10^{-3}$. The final loss is defined as
(21)$$\begin{array}{c} L=\alpha \cdot L_{d}+\left(1-\alpha \right)\cdot L_{s}, \end{array}$$
where α is balancing parameter. The selection of $N_{f\_s}$, $N_{rb\_s}$, and α is described in the experimental results section.

### 3.3. Integration into VVC Reference Software

#### 3.3.1. The Scope of the Proposed ABPN in VVC

In VVC, the QT-MTT block partitioning structure is applied to split a frame into various shapes and sizes of CUs. Based on CU-level, a prediction process is performed. Because applying the deep learning-based bi-prediction mode to all diverse cases of CUs increases computational complexity significantly, it is necessary to select only a few block types that greatly affect performance. Table 2 shows the number of CUs and area ratio for different CU size in the bi-prediction with the first 2 s on VTM-11.0 NNVC-1.0 anchor.

From Table 2, it can be seen that the area ratio with sizes 128×128, 64×64, and 32×32 accounts for 85.27%, 75.48%, 71.75%, and 59.23% on sequences *BQTerrace*, *BasketballDrive*, *PartyScene*, and *BQSquare*, respectively. Even though the percentages of the number of CUs of various sizes are evenly distributed, the percentage of pixel-based area is more important in terms of coding efficiency. Furthermore, as the resolution of the sequence increases, the role of a large square block becomes more crucial. From Table 2, the portions of area ratio with sizes 2N×N and N×2N seem relatively large. However, because these sizes contain 128×64, 64×128, 64×32, 32×64, 32×16, 16×32, 16×8, 8×16, 8×4, and 4×8 (all 10 block types), the percentages cannot be seen to be large. In accordance with the results in Table 2, the proposed deep learning-based bi-prediction mode replaces all of the traditional bi-prediction modes on CUs with sizes 128×128, 64×64, and 32×32. The proposed method is applied only on the luma component of the CU. The average mode is performed to the chroma component of the CU to which the proposed method is applied.

#### 3.3.2. The Strategy of the Proposed ABPN in VVC

The proposed ABPN is applied as a new bi-prediction mode in the existing motion-compensation process in VVC. Figure 4 shows the flow-chart of the bi-prediction in the motion-compensation process with the proposed method. First, the existence is checked for two reference prediction blocks. To execute an actual bi-prediction, both reference prediction blocks must exist. If only one of two exists, an existing block in itself is copied to the bi-prediction block buffer.

When both prediction blocks exist and the size of the CU is 128×128, 64×64, and 32×32, the deep bi-prediction mode with the proposed ABPN is performed. For other sizes of CU, the index of the BCW weight is checked. In BCW method, five weights are allowed, and one of these corresponds to the weight for the average mode. This average mode is called BCW_DEFAULT. The actual BCW mode is performed for the remaining four weights other than the BCW_DEFAULT mode. If BDOF mode is set to TRUE, averaging with BDOF-based refinement mode is performed. Otherwise, simple averaging mode is performed.

## 4. Experimental Results and Discussion

### 4.1. Generation of Training Dataset

To evaluate the proposed ABPN, the BVI-DVC [40] sequence dataset is used for generation of training dataset. The BVI-DVC dataset consists of 200 sequences at various resolutions 3840 × 2160, 1920 × 1080, 960 × 540, and 480 × 270. This dataset contains 800 sequences at four different resolutions. The BVI-DVC dataset covers a large variety of motion types, including camera motion, human actions, animal activity, etc. Therefore, it is suitable for establishing a training dataset. The VTM-11.0 NNVC-1.0 reference model is used to compress the sequences configured with random access (RA) under five quantization parameters (QPs) {22,27,32,37,42}.

In decoder of VTM anchor, for each CU size {32×32,64×64,128×128}, the L0 and L1 reference prediction blocks on bi-prediction process are extracted to utilize as the inputs of network. The corresponding ground-truth patches are cropped from raw video frames with the location of current CUs. For different block types {128×128,64×64,32×32} and QPs {22,27,32,37,42}, the dataset is constructed to train independently. Table 3 shows the number of pair of blocks for each type of training data. The dataset contains 8,972,496 pairs of blocks in total.

### 4.2. Training Settings

For the proposed ABPN, 15 models were trained with three CU sizes and five QPs. For training, we used the Adam optimizer [41] by setting $\beta_{1}=0.9$ and $\beta_{2}=0.99$. We adopted the cosine annealing scheme [42] and initially set the learning rate to $4\times 10^{-4}$. We trained with setting the size of mini-batch to 64. For each model, the number of iterations was set to 600 K. In other words, the number of epochs for each model is different as shown in Table 3. For all of the experiments, we observed that the weight converged to the optimal before-last epoch. The proposed ABPN was implemented in PyTorch on a PC with a Intel(R) Xeon(R) Gold 6256 CPU @ 3.60GHz and a NVIDIA Quadro RTX 8000-48GB GPU. We augmented the training data with random horizontal flips and 90° rotations.

### 4.3. KD-Based Training

To reduce the size of the deep neural network while maintaining the performance of the network, we applied the knowledge distillation (KD) [31] strategy in the training stage. In this paper, we constructed the architecture of the student model by setting the number of features of all CNN layers except the last layer and the number of residual blocks while keeping the framework the same as the teacher model.

For the selection of the two hyperparameters and alpha in loss function, we used the model with CU size 128×128 and QP = 37. First, we trained a teacher model with the number of features of all CNN layers except the last layer set to 64 and the number of residual blocks set to 10. The number of parameters of the teacher model is 1,295,169. Then, we trained the candidate student models by using pretrained teacher models on diverse settings of the number of features and the number of residual blocks and α value.

For hyperparameter determination, we constructed a validation dataset by using some of the JVET common test conditions (CTC) [13] sequences with 100 frames per sequence. We selected some sequences in all of the classes (*Tango2*, *CatRobot*, *BQTerrace*, *BQMall*, *BQSquare*, *ArenaOfValor*). The validation dataset contains 164,026 pair of blocks.

Figure 5 shows the number of parameters and peak signal-to-noise ratio (PSNR) with various hyperparameter settings on student models in KD-based training. As shown in Figure 5a, from the number of features 8 to 32, the PSNR increases steeply, and thereafter, it becomes smooth. Considering both the number of parameters of the model and the PSNR, we set the number of features of student model to 32. In the same manner, Figure 5b demonstrates that five residual blocks (RBs) are appropriate for a student model by considering the number of parameters and the quality of output. As shown in Figure 5c, after α value of 0.5, it is almost the same. Therefore, we selected 0.5 as the α value. As a result, the final model of the proposed ABPN consists of 32 features for each CNN layer except the last layer and five residual blocks. The number of parameters of the proposed ABPN is 231,905 in the lightweight network.

### 4.4. Encoding Configurations

The proposed ABPN has been integrated into VTM-11.0 NNVC-1.0 reference software. The PyTorch 1.8.1 was used for performing the proposed deep neural network-based bi-prediction mode in VTM. We have compared our scheme with VTM on two settings, VTM with bi-prediction baseline (only the average mode is used) and VTM anchor (all of the traditional bi-prediction modes are used). We follow the JVET common test conditions (CTC) for neural network-based video coding technology [13] in all experiments.

In the experiments, the low-delay B (LDB) and random access (RA) configurations are tested, under five QPs = {22, 27, 32, 37, 42}. At this time, sequences with the first 2 s for Class A to E are tested. The coding with the deep learning-based bi-prediction mode was conducted on the GPU in the same environment as training.

### 4.5. Comparisons with VVC Standard

The overall results of BD-rate [43] reduction are shown for the Y component because the bi-prediction mode with the proposed method is applied only on Y component. The running time ratios of encoding and decoding are indicated by EncT and DecT, respectively. The results of the BD-rate reduction and encoding/decoding computational complexity compared to VTM-11.0 NNVC-1.0 with bi-prediction baseline on RA and LDB are reported in Table 4. The proposed bi-prediction framework achieves significant BD-rate reductions on both RA and LDB configurations compared with traditional averaging bi-prediction method-based strategy. It is observed that 1.94% and 1.44% BD-rate savings can be achieved on average under RA and LDB, respectively. In particular, for *BQSquare*, the proposed bi-prediction method achieves up to 8.21% and 5.37% BD-rate reductions under RA and LDB, respectively. To describe the improvement of the proposed method in comparison with three bi-prediction modes (averaging, BCW, BDOF) of VVC, the results of the coding performance compared to VTM-11.0 NNVC-1.0 anchor are shown in Table 5. It is observed that the proposed method achieves up to 5.89% and 4.91% BD-rate savings under RA and LDB, respectively.

Although the whole encoding time increases highly, the overall decoding time has a little difference. Even though the proposed method adopted the CNN-based strategy, the proposed ABPN showed only on average 85% and 143% of decoding time under RA and LDB, respectively. From this result, we can deduce that the proposed integration strategy, which rarely changes the existing standard structure, is substantially effective in terms of decoding computational complexity. Video coding standards have specified only the format of the coded bitstream, syntax, and the operation of the decoder. In other words, the operation of the encoder is not a critical issue in video coding standards. Therefore, the experimental results show that the proposed bi-prediction method is suitable and offers a good trade-off between efficiency and complexity as compared to the existing VVC standard.

The qualitative results on *BQSquare* and *Johnny* are presented in Figure 6. The results of two sequences, the first column means the original frame and the second column means the compressed frame by VTM-11.0 NNVC-1.0 anchor. The last column means the compressed frame by applying the proposed ABPN-based bi-prediction.As shown in Figure 6, our approach is more robust in edge details than VTM anchor. It can be observed that the proposed method can reconstruct the signals similar to the original data.

Figure 7 describes the difference of block partition structures between traditional bi-prediction methods and the proposed ABPN-based bi-prediction method. Because the proposed bi-prediction mode is applied to three types of block {32×32, 64×64, 128×128}, CUs with these types increased. In particular, because the resolution of example in Figure 7 (416×240) is relatively small, CUs with 32×32 occupy the largest proportion. Furthermore, as the small size of CUs are merged into the large size of CUs, fewer coding bits are required in the overall video coding process. As a result, the proposed scheme can contribute to the whole coding efficiency.

### 4.6. Analysis on Performance of KD-Based Strategy and Attention Mechanism

To further analyze the contribution of the proposed ABPN, Table 6 shows the results of the ablation study for KD-based strategy and attention mechanism. We evaluated the model with CU size 128×128 and QP = 37 by using the validation dataset.

In order to analyze the computational complexity, Table 6 shows the number of model parameters and the number of floating-point operations (FLOPs) [44] for the teacher model of the proposed ABPN (ABPN-T), the proposed ABPN, and the proposed ABPN without KD strategy or attention mechanism, respectively. In this table, the gigaFlops (GFLOPs) denote a billion FLOPs. From Table 6, it can be seen that the difference between ABPN-T and ABPN is large in all the results of computational complexity measures. Furthermore, it is observed that the PSNR of the proposed ABPN is higher than the proposed ABPN without KD. Therefore, we can explain that the KD-based training strategy is helpful to obtain a model with high performance in lightweighting the deep learning model.

To train the proposed ABPN without attention mechanism, we used the concatenated tensor of two reference prediction blocks as an input of model. As with the proposed ABPN, the input tensor is fed into three CNN layers, two CNN layers, five residual blocks, and two CNN layers. Furthermore, as with the proposed ABPN, the KD-based training strategy is applied to train the model without an attention mechanism by using the pretrained teacher model. When comparing the proposed ABPN without attention, the proposed ABPN shows the higher PSNR than the model without attention. It demonstrates that the attention between two prediction blocks can extract more improved features. Therefore, in the fusion of two images, it is more effective to apply a complicated structure, such as an attention mechanism, than to use general CNN layers only.

Figure 8 shows the visualization of output feature map of CNN layer for the proposed ABPN. We used the 128×128 prediction blocks in POC20 of the *BQTerrace* under RA and QP = 37 as the inputs. We extracted the first four output feature maps in all CNN layers. In Figure 8a,b, the upper row means a ground truth (GT) block. The GT block of Figure 8a contains the ripples. The GT block of Figure 8b contains some chairs, tables, and a person.

We visualized the output feature maps of each three CNN layers for the first prediction block before the designed attention module. In addition, we selected a CNN layer between attention module and residual blocks to prove the effect of attention. Finally, we selected a previous CNN layer of the last layer. For all of the results, the feature map becomes stronger from the top layer to bottom layer. In particular, the results demonstrate that the finer details can be recovered after the attention module. Therefore, the proposed ABPN is able to generate the enriched features by applying the designed attention mechanism.

## 5. Conclusions

In this paper, we have proposed an attention-based bi-prediction network (ABPN) to effectively improve the performance of bi-prediction in VVC. The proposed bi-prediction method is able to handle various kinds of motion variations in a nonlinear mapping manner. The structure of the proposed ABPN consists of the attention mechanism, and the local and global skip-connection. With this structure, it can generate a precise fused feature. In addition, by adopting the knowledge distillation (KD)-based training strategy, we have reduced the number of parameters of the network considerably. The proposed ABPN is integrated into VVC as a novel bi-prediction method. Experimental results demonstrate that the proposed ABPN can significantly enhance the overall coding performance.

Compared to the VTM-11.0 NNVC-1.0 that uses only the averaging mode, the proposed method yielded 1.94% and 1.44% BD-rate reductions on average for the Y component under RA and LDB, respectively. The proposed ABPN achieved 0.86% and 1.37% BD-rate savings on average for the Y component under RA and LDB, respectively, compared with the VTM-11.0 NNVC-1.0 anchor, which uses all of the bi-prediction modes. As a consequence, the proposed method can improve the effectiveness of video transmission scheme for video sensor network over the 5G network.

In this work, the bi-prediction block was generated by the proposed ABPN. In future work, the inter prediction scheme can be further improved by enhancing uni-prediction block. In addition, an extending training dataset with additional encoding information, such as merge mode flag, can give a better coding performance.

## Figures and Tables

**Figure 1 sensors-23-02631-f001:**
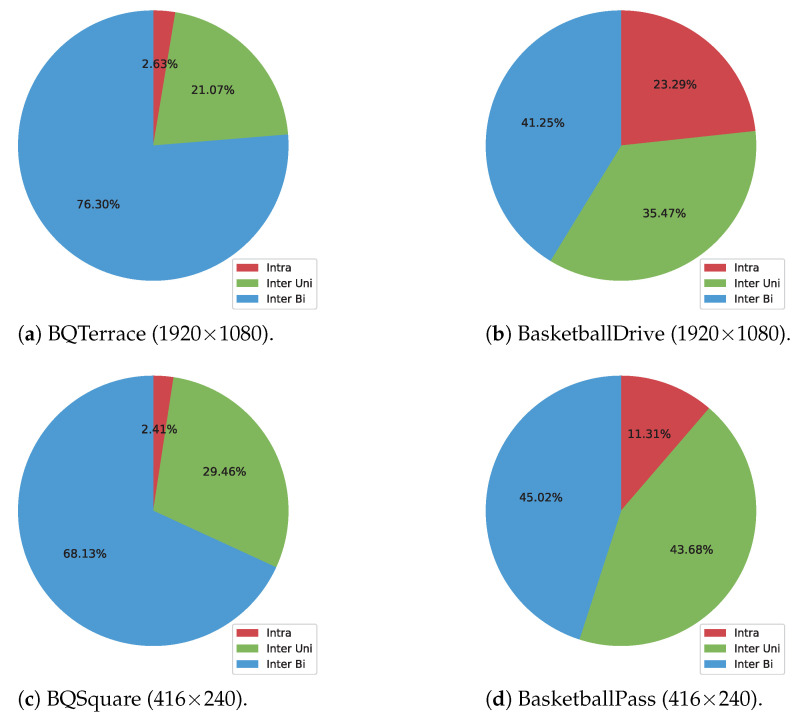
The ratio of number of CUs for selected modes with the first 2 s results under random access (RA) and QP37.

**Figure 2 sensors-23-02631-f002:**
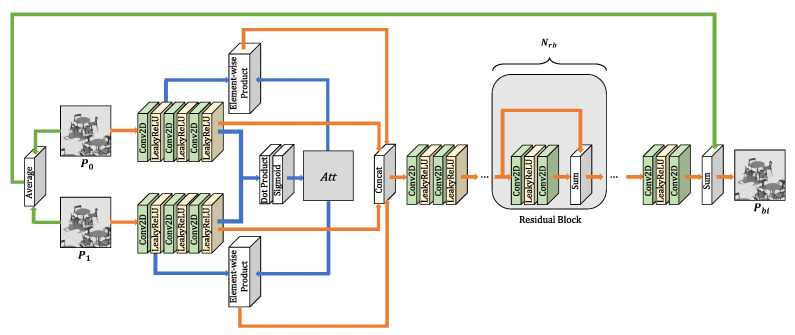
An overview of the proposed network.

**Figure 3 sensors-23-02631-f003:**
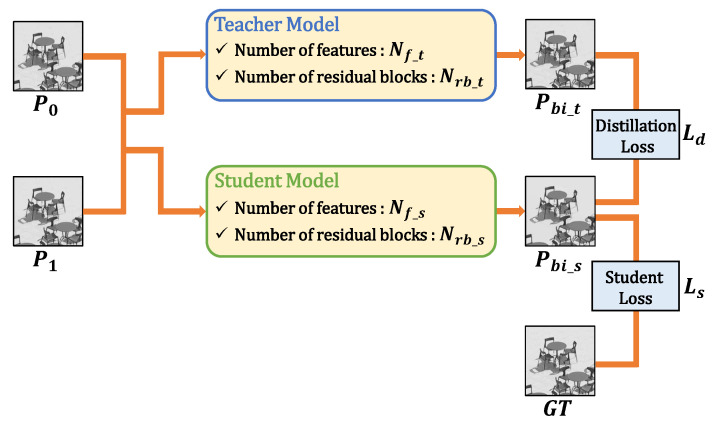
The structure with knowledge distillation (KD)-based learning strategy.

**Figure 4 sensors-23-02631-f004:**
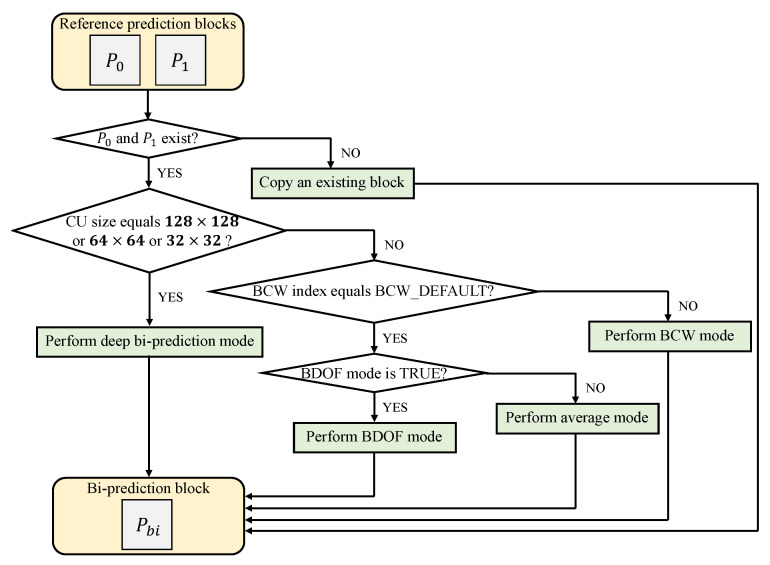
The flowchart of the bi-prediction in the motion compensation process with the proposed method in VVC.

**Figure 5 sensors-23-02631-f005:**
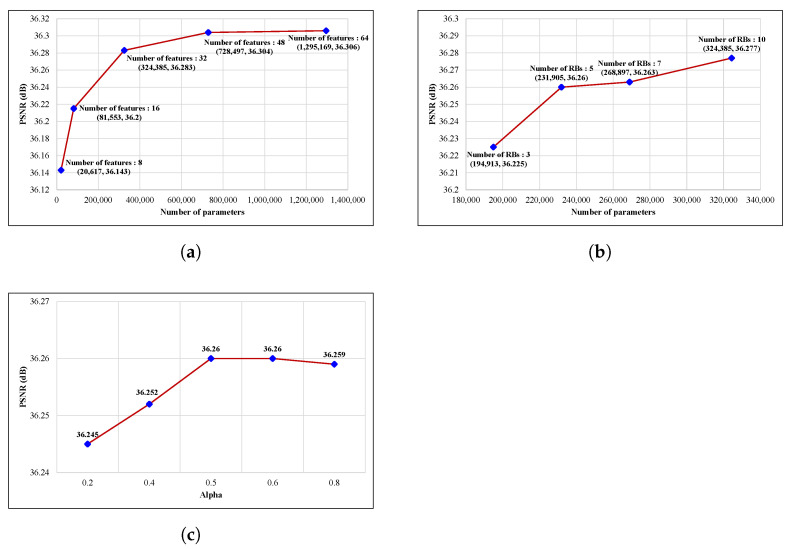
The number of parameters and PSNR with various hyperparameter settings on student models in KD-based training. (**a**) Comparison on number of features with 10 residual blocks and α= 0.5. (**b**) Comparison on number of residual blocks with 32 features and α= 0.5. (**c**) Comparison on α with 32 features and 5 residual blocks.

**Figure 6 sensors-23-02631-f006:**
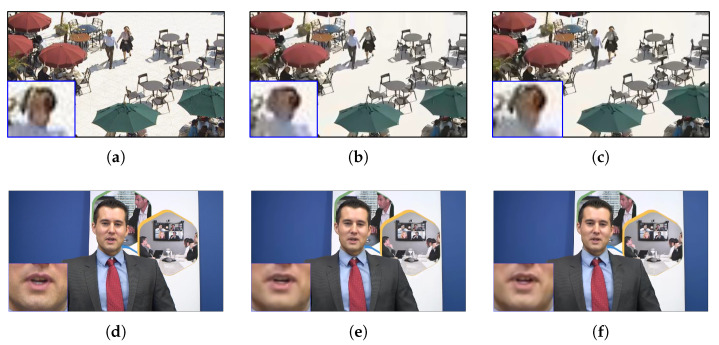
Visual comparison between VTM-11.0 NNVC-1.0 anchor and the proposed ABPN-based bi-prediction. Top: POC116 of *BQSquare* under LDB and QP37. Bottom: POC48 of *Johnny* under LDB and QP37. (**a**) Original frame of *BQSquare*. (**b**) VTM-11.0 NNVC-1.0 anchor (Y PSNR 28.465 dB). (**c**) ABPN-based bi-prediction (Y PSNR 28.505 dB). (**d**) Original frame of *Johnny*. (**e**) VTM-11.0 NNVC-1.0 anchor (Y PSNR 37.935 dB). (**f**) ABPN-based bi-prediction (Y PSNR 38.065 dB).

**Figure 7 sensors-23-02631-f007:**
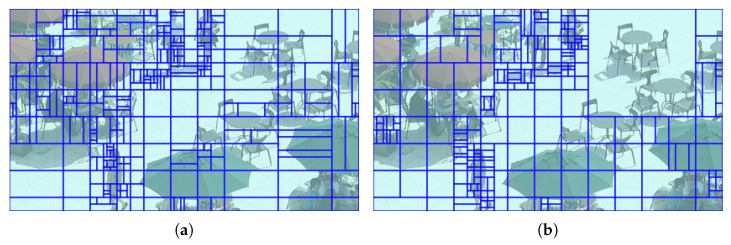
Illustration of the block partition structure of VTM-11.0 NNVC-1.0 anchor and the proposed ABPN-based bi-prediction. POC74 of *BQSquare* under LDB and QP27. (**a**) VTM-11.0 NNVC-1.0 anchor (5992 bits/Y PSNR 34.8423 dB). (**b**) ABPN-based bi-prediction (4848 bits/Y PSNR 34.8509 dB).

**Figure 8 sensors-23-02631-f008:**
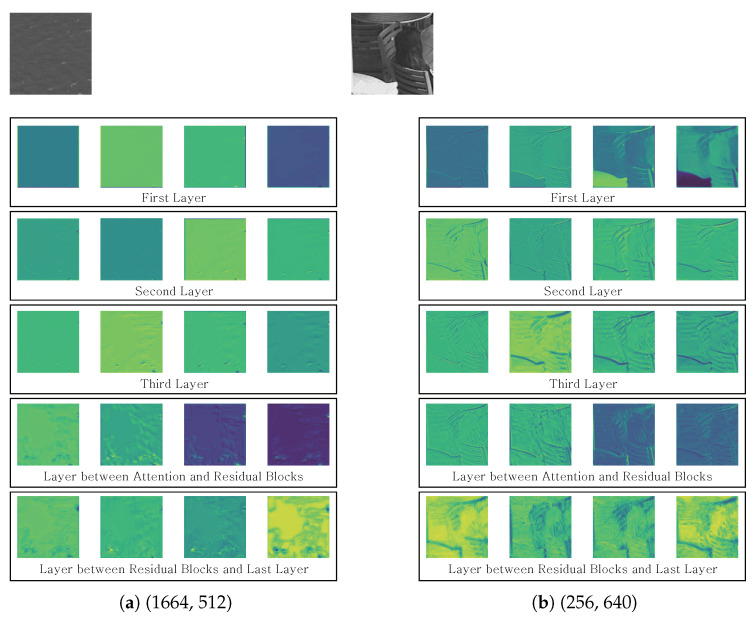
Visualization of output feature maps of CNN layers. Input: 128×128 prediction blocks in POC20 of *BQTerrace* under RA and QP37. (x,y) means the top left position of CU in frame.

**Table 1 sensors-23-02631-t001:** Problem formulation and related methods for deep learning-based inter prediction in video coding.

Categories	Problem Formulation	Methods	Coding Standards
Uni-prediction block enhancement	Refine the uni-prediction block	CNNMCR [32]	HEVC
		NNIP [33]	
Bi-prediction block generation	Generate the bi-prediction block	CNN-based bi-prediction [28]	
		STCNN [29]	
Frame extrapolation and interpolation	Estimate additional reference frame from existing reference frames	VECNN [34]	
		MQ-FKCNN [35]	HEVC and VVC
		Deep network-based frame extrapolation [36]	
		Affine transformation-based deep frame prediction [37]	HEVC

**Table 2 sensors-23-02631-t002:** The number of CUs and area ratio for different CU size in bi-prediction under RA and QP37.

	Sequence			
	BQTerrace (1920 × 1080, B Class)		BasketballDrive (1920 × 1080, B class)	
CU size	Number of CUs	Area ratio	Number of CUs	Area ratio
128 × 128	28.22%	78.58%	13.28%	59.96%
64 × 64	8.25%	5.75%	11.15%	12.59%
32 × 32	5.38%	0.94%	10.37%	2.93%
16 × 16	2.68%	0.12%	3.45%	0.24%
8 × 8	0.81%	0.01%	0.38%	0.01%
2N × N, N × 2N	27.15%	11.53%	33.03%	18.96%
Others	27.50%	3.08%	28.33%	5.32%
	Sequence			
	PartyScene (832 × 480, C class)		BQSquare (416 × 240, D class)	
CU size	Number of CUs	Area ratio	Number of CUs	Area ratio
128 × 128	3.08%	32.57%	3.62%	26.59%
64 × 64	11.52%	30.48%	15.34%	28.19%
32 × 32	13.16%	8.70%	9.68%	4.45%
16 × 16	8.15%	1.35%	4.85%	0.56%
8 × 8	3.92%	0.16%	2.16%	0.06%
2N × N, N × 2N	36.05%	20.90%	39.93%	35.59%
Others	24.13%	5.84%	24.43%	7.56%

**Table 3 sensors-23-02631-t003:** The number of pair of blocks for each type of training data.

CU Size	QP	Number of Pair of Blocks	Number of Epochs (Number of Iterations = 600,000 and Batch Size = 64)
128×128	22	549,936	70
	27	641,210	60
	32	501,868	77
	37	237,373	162
	42	174,378	221
64×64	22	1,500,840	26
	27	767,004	51
	32	424,074	91
	37	206,211	187
	42	141,052	273
32×32	22	2,351,966	17
	27	866,121	45
	32	376,900	102
	37	159,092	242
	42	74,471	516
Overall		8,972,496	

**Table 4 sensors-23-02631-t004:** BD-rate reduction and encoding/decoding computational complexity compared to VTM-11.0 NNVC-1.0 baseline.

Class	Sequence	RA			LDB		
		BD-Rate	EncT	DecT	BD-Rate	EncT	DecT
Class A1 (3840 × 2160)	Tango2	−0.97%	2494%	108%	−0.42%	2889%	504%
	FoodMarket4	−0.89%	2641%	235%	−1.58%	3558%	635%
	**Average**	**−0.93%**	**2568%**	**172%**	**−1.00%**	**3224%**	**569%**
Class A2 (3840 × 2160)	CatRobot	−2.02%	2523%	160%	−1.73%	2153%	224%
	DaylightRoad2	−2.04%	2627%	198%	−1.39%	2211%	370%
	**Average**	**−2.03%**	**2575%**	**179%**	**−1.56%**	**2182%**	**297%**
Class B (1920 × 1080)	MarketPlace	−1.93%	2035%	147%	−1.13%	2204%	227%
	RitualDance	−1.00%	1481%	79%	−0.54%	1727%	234%
	Cactus	−1.64%	2334%	173%	−1.70%	2504%	136%
	BasketballDrive	−1.42%	1819%	191%	−0.78%	2675%	160%
	BQTerrace	−2.50%	3177%	128%	−2.14%	4795%	127%
	**Average**	**−1.70%**	**2169%**	**144%**	**−1.26%**	**2781%**	**177%**
Class C (832 × 480)	BasketballDrill	−1.12%	2882%	133%	−0.86%	2327%	232%
	BQMall	−1.86%	2563%	156%	−1.19%	2000%	191%
	PartyScene	−1.70%	2645%	327%	−1.38%	2312%	523%
	RaceHorses	−0.66%	1076%	84%	−0.18%	769%	98%
	**Average**	**−1.34%**	**2291%**	**175%**	**−0.90%**	**1852%**	**261%**
Class D (416 × 240)	BasketballPass	−2.15%	1414%	111%	−0.46%	972%	126%
	BQSquare	−8.21%	2769%	430%	−5.37%	2551%	510%
	BlowingBubbles	−1.83%	1839%	224%	−0.77%	1075%	191%
	RaceHorses	−1.09%	917%	172%	0.17%	557%	188%
	**Average**	**−3.32%**	**1735%**	**234%**	**−1.61%**	**1289%**	**254%**
Class E (1280 × 720)	FourPeople	-	-	-	-2.63%	2302%	160%
	Johnny	-	-	-	-2.63%	2246%	97%
	KristenAndSara	-	-	-	-2.06%	2113%	82%
	**Average**	**-**	**-**	**-**	**−2.44%**	**2220%**	**113%**
All Sequences	**Overall**	**−1.94%**	**2190%**	**180%**	**−1.44%**	**2197%**	**251%**

**Table 5 sensors-23-02631-t005:** BD-rate reduction and encoding/decoding computational complexity compared to VTM-11.0 NNVC-1.0 anchor.

Class	Sequence	RA			LDB		
		BD-Rate	EncT	DecT	BD-Rate	EncT	DecT
Class A1 (3840 × 2160)	Tango2	−0.17%	2455%	147%	−0.38%	2636%	453%
	FoodMarket4	-0.57%	3508%	314%	−1.42%	3652%	718%
	**Average**	**−0.37%**	**2981%**	**230%**	**−0.90%**	**3144%**	**586%**
Class A2 (3840 × 2160)	CatRobot	−1.14%	2803%	174%	−1.57%	1680%	245%
	DaylightRoad2	−0.85%	2426%	218%	−1.48%	1971%	362%
	**Average**	**−0.99%**	**2615%**	**196%**	**−1.53%**	**1826%**	**304%**
Class B (1920 × 1080)	MarketPlace	−0.63%	2418%	151%	−1.21%	1931%	226%
	RitualDance	−0.11%	1266%	78%	−0.63%	1638%	223%
	Cactus	−0.76%	1818%	172%	−1.63%	2150%	138%
	BasketballDrive	−0.35%	1572%	204%	−0.70%	2568%	170%
	BQTerrace	−1.17%	3650%	132%	−1.33%	4121%	126%
	**Average**	**−0.60%**	**2145%**	**147%**	**−1.10%**	**2482%**	**177%**
Class C (832 × 480)	BasketballDrill	−0.51%	2007%	121%	−0.86%	2311%	167%
	BQMall	−0.48%	2025%	177%	−1.22%	2002%	196%
	PartyScene	−0.85%	1548%	194%	−1.39%	2298%	411%
	RaceHorses	−0.12%	776%	89%	−0.02%	832%	99%
	**Average**	**−0.49%**	**1589%**	**145%**	**−0.87%**	**1861%**	**218%**
Class D (416 × 240)	BasketballPass	−0.45%	1423%	106%	−0.55%	1126%	128%
	BQSquare	−5.89%	2538%	454%	−4.91%	2783%	484%
	BlowingBubbles	−0.50%	1783%	248%	−0.78%	1062%	185%
	RaceHorses	−0.13%	835%	172%	−0.09%	483%	175%
	**Average**	**−1.74%**	**1645%**	**245%**	**−1.58%**	**1364%**	**243%**
Class E (1280 × 720)	FourPeople	-	-	-	−2.43%	2423%	163%
	Johnny	-	-	-	−2.57%	2753%	100%
	KristenAndSara	-	-	-	−2.13%	2086%	81%
	**Average**	**-**	**-**	**-**	**−2.38%**	**2421%**	**115%**
All Sequences	**Overall**	**−0.86%**	**2050%**	**185%**	**−1.37%**	**2125%**	**243%**

**Table 6 sensors-23-02631-t006:** The performance comparison between the teacher model of ABPN, lightweighted ABPN, and ABPN without KD strategy or attention mechanism.

	PSNR (dB)	Params.	GFLOPs
**ABPN-T**	36.306	1,295,169	10.615
**ABPN**	36.260	231,905	1.900
**ABPN without KD**	36.227	231,905	1.900
**ABPN without Attention**	36.221	139,617	1.150

## Data Availability

Not applicable.

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
