# Peer review of "Attention-Based Bi-Prediction Network for Versatile Video Coding (VVC) over 5G Network"

_sensors, 2023, doi:10.3390/s23052631_

Round 1

Reviewer 1 Report

1.The authors have proposed an attention-based bi-prediction network (ABPN) to effectively improve the performance of bi-prediction in Versatile Video Coding (VVC). The proposed ABPN is integrated into VVC as a novel bi-prediction method.

2.The proposed ABPN is designed to learn efficient representations of the fused features by utilizing attention mechanism. The proposed bi-prediction method is able to handle various kinds of motion variations in non-linear mapping manner.

3.The part of experimental results is presented at an excellent level. The dataset for the experiments covers a large variety of motion types, including camera motion, human actions, animal activity, etc. The experimental results demonstrate that the proposed ABPN can significantly enhance the overall coding performance.

4.In section ‘Related works’ it may be useful to add a structured description of close methods, for example: (i) a table of the close methods, (ii) a special figure (e.g., with a framework, with a taxonomy of the close methods).

5.It may be interesting to point out in the conclusion some prospective future research direction(s).

6.In general, the paper is prepared at a very good level (all part) and can be accepted.

Author Response

Dear reviewer,

Thank you for your valuable comments. Please, refer to the attached replies.

Reviewer 2 Report

In this paper, the authors propose an attention-based bi-prediction network to enhance the quality of bi-prediction block by using a CNN-based manner. However, some aspects were unclear to the reviewers:

1.     The method of using convolutional neural network and knowledge distillation can improve the coding efficiency, but the computational complexity introduced cannot be ignored. Have you considered the trade-off between efficiency and complexity?

2.     In the statistics of different CU sizes in Table 1, the author's paper classifies rectangular CU into one category, but in fact, the proportion of rectangular CU of some sizes may be higher than that of square CU, which seems to be invisible from Table 1.

3.     I suggest the authors to put more result images/samples for better demonstration. Also please give more details about the model parameters, along with training parameters. Please clarify how often should the training of the deep networks be repeated.

4.     Please justify the proper choice of sigmoid and LeakyReLU activation functions.

5.     From your experiment, we can see that it is compared with VTM 11.0, which seems to lack some comparative experiments. In addition, VTM has been updated to 19.0. Why not compare with the latest version?

Author Response

Dear Reviewer,

Thank you for your valuable comments. Please, refer to the attached replies.

Reviewer 3 Report

This paper proposes an attention-based bi-prediction network for Versatile Video Coding. The knowledge distillation (KD)-based training strategy is adopted to reduce the number of network parameters. The proposed innovation points have been proved in the experiment, and the research has some reference value for video transmission over 5G network. However, there are still some problems as follows:

1. The uses of formula punctuation are ambiguous, partly using commas and partly using periods. It can be deleted or unified.

2. In Figure 2, the proposed network diagram needs to be improved. The final bi-prediction block consists of the addition of two parts, arrow pointing need to be added to the green line, and the position of sum block should be adjusted.

3. The format of the reference papers is not uniform. For example, the font is sometimes not consistent, the page number is sometimes not marked (e.g.13,22,27,31,40,41,42,43,44), sometimes "pp" is used (e.g.1,14,15,17,32,33,38), and sometimes the length of the line of the expansion is not consistent (e.g.14).

4. In the experimental part, compared with VTM-11.0 NNVC-1.0 baseline and anchor, the proposed network showed better performance at BD-rate, but the running time ratio of encoding and decoding is large, which should be explained.

Author Response

(The authors gave the same response as above.)

Round 2

Reviewer 2 Report

  • The author answered my question very well, and the research of the paper was accepted.